# From State Control to Regulation to Privatization of Religion–State Relations in Israel: *Kashrut* Reform as a Case-Study

Nahshon Perez *  and Elisheva Rosman *

Department of Political Studies, Bar Ilan University, Ramat Gan 5290002, Israel
* Correspondence: perezna@biu.ac.il (N.P.); rosmane@biu.ac.il (E.R.)

**Abstract:** Religion–state relations in Israel have been defined as following the status-quo agreement. This agreement, going back to the founding of Israel, allows recognized religious groups a monopoly regarding issues of personal status, and promises religious goods and exemptions to such groups (mainly, but not limited, to Orthodox Judaism). Since the mid 1980s, Israel has changed its economic policies, from a centralized economy to a privatized, liberalized system. This economic change introduced significant shifts within Israeli society. These include major recent changes in religion–state relations, most importantly the reform in *kashrut* certification, and growing commercial activity during the Sabbath. Such changes demonstrate a dynamic of state retreat, from a direct statist provision of religious goods, to the state either retreating completely, or re-situating itself as a regulatory organ. Using the *kashrut* reform as a case study, we suggest that the status-quo model can no longer adequately define religion–state relations in Israel, and is being replaced by a hybrid model, which includes libertarian, regulation-based, and the noted status quo attributes. We conclude with noting the significance of this development for the Jewish character of Israel.

**Keywords:** *kashrut*; Israel; status-quo; religion–state relations; liberal economic reform

## 1. Introduction

In 2021 the Israeli parliament radically changed its long-standing policy regarding *kashrut* rules. This change reduced governmental involvement in a central aspect of religion–state institutions in the state of Israel. In this article, we argue that this policy, important and noteworthy in itself, should be seen as a part of a larger change in religion–state relations in Israel. While it is customary to analyze religion–state relations in Israel through the long-standing status quo model, we shall argue that it would be more accurate to describe the emerging model of religion–state relations in Israel as hybrid, with parts of the status quo model still intact, others regulation-based, and others increasingly or completely privatized. The suggested hybrid model of religion–state relations, we shall argue, better captures recent events in religion–state relations, and provides a framework for analysis that enables the classification and exploration of future cases. In order to demonstrate this suggested hybrid framework, we shall point to the recent (2021) *kashrut* reform.[1] The language used within this new law is economic-based, making it an appropriate case study for examining the shift in institutional thinking regarding religion–state relations in Israel. As stated in preliminary comments on the 'economic plan law', the policy prior to the reform "creates a monopoly framework, negatively influencing the cost of living, as it prevents competition in the field of *kashrut*, and does not allow coordinating *kashrut* provision with the various wishes of consumers interested in *kashrut* keeping".[2] In other words, the *kashrut* reform in Israel demonstrates how even in a religious sphere, where the State is heavily invested, economic considerations can be influential in shifting the religion–state arrangements.[3] This case study, therefore, can serve as a hypothesis-generating case (Gerring 2004) that can

shed light on the field and enhance our understanding of economic considerations within religion–state institutions.[4]

This article is structured as follows: Section 2 briefly presents the model of religion–state relations in Israel, generally known as the 'status-quo' model; Section 3 explains the recent *kashrut* reform plan; and Section 4 places the *kashrut* reform plan in a more general context of economic liberalization of the Israeli economy. The final section suggests that rather than viewing the *kashrut* reform as an anomaly to the status quo model, it should be seen as a foreseeable continuation of the laissez faire economic framework of Israel. We conclude by suggesting that religion–state relations in Israel should be viewed as a hybrid model, with status quo, regulatory, and privatized elements.[5]

## 2. The Model of Religion–State Relations in Israel

Religion–state relations in Israel originated, by and large, in the Ottoman millet system. In this system, religious denominations were recognized by the state[6] and enjoyed exclusive jurisdiction over issues of personal law (exemplifying this historical connection is the lack of civil marriages in Israel to the present day). In the Israeli version of the millet system, the arrangement between religion and state is based on a series of laws and regulations usually referred to collectively as the 'status quo compromise' or agreement (Barak-Erez 2008; Liebman and Don Yihya 1983; Neuberger 1999; Cohen and Kampinsky 2006).

This agreement goes back to the period just prior to the ending of the British Mandate. As noted by Barak-Erez, this agreement was solidified in "the letter that the Jewish Agency—the main Zionist institution at the time (which was controlled by the secular Labor Party)—sent in 1947 to the international organization of Agudat Israel, the hegemonic movement within the ultra-Orthodox Jewish public. The letter, known as the "status quo document," included commitments to observe certain traditions in the future state . . . the recognition of the Jewish Sabbath (Saturday) as the official day of rest; the provision of kosher food in public institutions; the exclusivity of the religious law of marriage and divorce; and a commitment to ensure the autonomy of the ultra-Orthodox educational system." (Barak-Erez 2008, pp. 2496–97).

This organizing principle of religion–state relations in Israel still partly exists: in the four spheres of personal status (the autonomy of various recognized religious communities over marriage and divorce of their members), observance of the Sabbath, *kashrut* super-vision, and the autonomy of ultra-orthodox Jews regarding education. Importantly, the principle of safeguarding the autonomy of religious communities is accompanied with substantial state funding, including salaries paid to clergy, funding of religious schools, maintaining and funding of religious courts, and more. However, this organizing framework faces fierce criticism (Cohen and Susser 2000; Ben-Porat 2013). It also seems to be undergoing a slow, at times almost unnoticeable, process of privatization and change, as described below.

Two crucial aspects of the status-quo agreement, the religious monopoly over issues of personal law and the autonomy of ultra-orthodox communities, require a brief explanation. The status quo creates phenomena one does not expect to see in a democratic state. This is perhaps seen most clearly in the religious monopoly over issues of personal law. Civil marriages do not exist in Israel and Israeli citizens of any religion cannot marry without the official sanction of their respective 'religions'. This also renders interreligious marriages impossible and creates the situation where if one is an atheist, or unattached to any religion, one must still submit to a religious authority in order to marry or divorce.[7] For example, Jews must marry via the Chief Rabbinate, a state organ, delegated to Orthodox authority, even if they are not religious or not orthodox.

At the same time, a couple married outside of Israel in a civil ceremony can return to Israel and register as married; the state will recognize these marriages provided they took place in another country. This is an ad-hoc 'solution' for couples who are either unwilling or unable to marry via the Rabbinate. It allows those who reject the option of a religious marriage to still be recognized by the State as married for all legal purposes, and also allows

those who are unable to be married through the religious establishment (interreligious couples, individuals who are not recognized as belonging to any religion, LGBT couples, as well as others) to be recognized by the State as married.

Furthermore, couples are not required to register as married in order to receive State recognition as a family. Maintaining a joint household, the father claiming paternity, are all enough to qualify for de facto recognition. In the past decade, more couples are opting to marry privately (mostly via religious ceremonies) without registering officially, but recognized as married by their social circles (and enjoy all statist benefits accorded to married couples), and—de facto—by the state (Halperin-Kaddari et al., forthcoming).

In other words, while the State is heavily invested in what it sees as the religious sphere, it is willing to recognize certain loopholes in order to not (completely) infringe on the civil rights of citizens who cannot marry through this framework.

Moving to another controversial aspect of the status-quo agreement in Israel, the Ultra-Orthodox (*haredi*) community in Israel is allowed certain rights and exemptions the majority of Jewish and non-Jewish Israelis are not granted (Berman 2000; Perez 2014; Golan-Nadir 2022). For example, the haredi educational system enjoys significant exemptions from the curriculum taught at public schools in Israel, including sciences, English, and mathematics (Friedman 1991). Subjects taught at Ultra-Orthodox schools are almost exclusively religious, especially for boys, while girls' studies include a larger measure of 'general' studies. This is despite the fact that the haredi system of education is partially state-funded.[8] Another exemption enjoyed by Ultra-Orthodox Jews is that they are granted long-term deferrals from military service (that usually result in exemptions), despite the fact that conscription is mandatory for other Jewish citizens.[9]

While accommodation of religious minorities is a liberal virtue that is not unique to Israel, the relationship between the State and the haredi community extends beyond usual accommodations. Allocation of resources in religious affairs, while egalitarian in principle, is biased towards Orthodox Judaism.

It is important to note that the Jewish aspects of personal law regulation are duplicated for non-Jewish denominations and communities, with the state allowing various Christian, Muslim, and Druze authorities autonomy regarding personal law (Karayanni 2020). Since this article focuses on the status of the majority in relation to the economic process the state is undergoing, its scope does not allow for the exploration of these minorities' situation.

It goes without saying that the status-quo system faced, and still faces, various challenges and fierce critiques, inclusive of court petitions, political battles, and social protests. The scope of this article prohibits us from presenting a comprehensive review of such struggles (see Sapir and Statman 2019). The aspects of the status-quo agreement that are arguably most controversial in Israel are the lack of civil marriages and the Ultra-Orthodox unique education system. Arguably, the most problematic is the Ultra-Orthodox education system, which creates a reality whereby its graduates lack appropriate skills for a modern labor market. As the percentage of Ultra-Orthodox Jews grows, this specific outcome of the status-quo agreement is perceived by major Israeli economists as a threat to the entire Israeli economy (Zeira 2021, pp. 310–17; Ben-David 2022).[10]

Now that the Israeli religion–state framework has been explained, we may turn to the *Kashrut* reform initiated by the Minister for Religious Affairs, Matan Kahana.

## 3. The Proposed *Kashrut* Reform

This section presents an overview of the *Kashrut* reform. As the topic is complex and comprises of many details, it is divided into four sub-sections: a brief explanation regarding the meaning of *kashrut*, (Section 3.1); explaining the policy preceding the reform (Section 3.2); the reform itself (Section 3.3); and the responses to the reform (Section 3.4).

### 3.1. The Meaning and Context of Kashrut

Jewish Law (*halakha*) dictates strict dietary restrictions. These include the separation of milk and meat products, the prohibition against consuming certain animals and foodstuffs,

the inspection of vegetables and grain for insects, as well as additional prohibitions pertaining to the person doing the actual cooking or baking and the handling of certain ingredients, including wine (see for example: Shulkhan Aruch, Yoreh De'ah, 1–134). Termed the laws of *Kashrut*, these edicts are the basis for Jewish life, along with the issues pertaining to the keeping of the Sabbath (*Shabbat*) and the laws of family purity (*taharat haMishpacha*). These three categories of halakha are seen as the main axis of Jewish Orthodox life. Many Jews who are not strictly Orthodox also adhere to the laws of *Kashrut*, to varying extents, and even those who do not necessarily adhere to the rules of *Shabbat* frequently attach importance to certain aspects of *Kashrut*, for example, abstaining from pork. In order to allow those who observe *Kashrut* to buy permitted products and to eat outside the home, often Jewish communities around the world supervise the manufacturing of food products, as well as restaurants and catering services, in their vicinity. Israel is no exception: 70% of Jewish Israelis keep kosher outside of their homes, with an even larger percentage doing so within their homes (Friedman and Finkelstein 2017). Consequently, there is an expectation that the State, seeing itself as a Jewish state, will do no less and will be involved in the supervision of *kashrut*.

### 3.2. The Pre-Reform Policy

Due to *kashrut*'s status as a main staple of Jewish living, the Rabbinate (a state organ) has always overseen the maintenance of *kashrut* in the public sphere in Israel, as a part of the status-quo agreement. Religious councils,[11] as organs of the Rabbinate, are the only authority allowed to award "*kashrut* certificates" to manufacturers, catering services, restaurants, and so on. Since its inception, the Rabbinate has always maintained this monopoly, not allowing any competition from other religious authorities (Friedman and Finkelstein 2017).

While obtaining a *kashrut* certificate is voluntary, in practice this is not so clear-cut. A catering service or hotel which does not have certification cannot apply for government tenders and is limited in the clientele they can serve. Thus, obtaining certification in Israel is almost imperative for many suppliers (Filber 2018).

That said, it is debatable whether the *kashrut* system in Israel is a "pure" monopoly. A monopoly is defined as "the exclusive possession or control of the supply of or trade in a commodity or service" (Oxford dictionary). In its current form, while it is not possible to give a certificate using the word "kosher" without Rabbinate sanction, alternative certificates—that do not use the word "kosher", but use comparable wordings—exist (Koren 2017), thereby creating a situation where one can be perceived as "kosher" to some clientele even without Rabbinate sanction.

Likewise, the Rabbinate does not have full control over its own rabbis who may award *kashrut* certificates. According to the pre-reform law, the Rabbinate ordains rabbis who are then able to award *kashrut* certificates. It should be noted that the Rabbinate serves as a gatekeeper in the sense that only Orthodox rabbis who pass its set exams can then work in municipal or regional councils and award *kashrut* certificates. However, once recognized as rabbis by the Rabbinate, rabbis can (and do) make autonomous decisions regarding halakhic issues. In practice, while the Rabbinate provides an official "umbrella" for the system, each local religious council is in charge of its own *kashrut* system. The local rabbi of a given town, city, or regional council has the ability to determine what constitutes "kosher" in his halakhic opinion. He can then decide what places of business must do in order to be eligible for *kashrut* certification; being more stringent or more lenient in his rulings as he wishes, without any oversight. In other words, while the rabbi himself receives his salary from the state and is officially an employee of the Rabbinate, he is autonomous in deciding *kashrut* criteria. There are no universal criteria decided by the Rabbinate (Friedman and Finkelstein 2017; Filber 2018).

Further to this point, the haredi community maintains parallel *kashrut* authorities, which usually provide *kashrut* supervision in addition to that given by the Rabbinate (as opposed to the alternative *kashrut* organizations). This creates a reality where those who

cater to a haredi clientele have to pay for double supervision: Rabbinate *Kashrut* and haredi (seen as more stringent) supervision. As noted above, only the Rabbinate is allowed to issue a certificate using the term "kosher". Therefore, only once one has a certification from the Rabbinate can one acquire additional (haredi) supervision. That said, since alternative *Kashrut* agencies have entered the market, one can obtain a certificate which does not use the word "kosher", but is recognized by many consumers as sufficient (Koren 2017).

Moreover, Rabbinate supervision allows for corruption: the *Kashrut* supervisor, although an employee of a given religious council, is usually paid directly by the owner of the establishment he supervises.[12] The supervisor arrives at the facility to be supervised, and is paid on the spot for his work. This is known to have caused some supervisors to invent additional restrictions and claim they were imperative to *kashrut*, or to hint that, if given particular benefits, they would be inclined to overlook certain *kashrut* problems (Ilan 2021a; Houminer 2021). Likewise, since the owner of the establishment paid the *Kashrut* supervisor's salary directly, this could lead to the opposite situation where the supervisor felt obligated to ignore certain *Kashrut* standards or else not be paid (Friedman and Finkelstein 2017).

In general, the maintaining of *kashrut* supervision is costly. As noted, those who wish to have both Rabbinate and haredi supervision in order to boost their clientele are required to pay even more. The Ministry for Religious Affairs regulates the fees for *kashrut* certification. The fee is determined by the size of the establishment being supervised (how many people it can serve, how large it is, how many employees it has, and so on, all depending on the type of establishment).[13] These can range from approximately 500 NIS per year to almost 10,000 NIS per year, not including Passover (which adds various layers of complexity beyond our subject matter). Adding another supervising authority can double the price, if not more. Haredi supervision is not regulated in any way, and the certifying rabbi is autonomous in deciding fees and criteria for supervision.

Perhaps unsurprisingly, the Israeli press has often discovered that Rabbinate supervision was substandard, if not blatantly non-existent (Friedman and Finkelstein 2017), justifying those who turned to haredi supervision. The State Comptroller has specified the pre-reform situation as unsatisfactory, to say the least (State Comptroller Report (Israel) 2017). All of these problems raised severe doubts regarding the Rabbinate's Kashrut supervision system and emphasized the need for reform.

During the past decade, a number of attempts to establish independent *Kashrut* supervision institutions were made. The most famous of these was *Hashgacha Pratit* ("private supervision" in Hebrew, which is also a play on the term for personal divine providence), and later, the Tzohar[14] *Kashrut* initiative. The Rabbinate waged aggressive campaigns against these attempts, succeeding in outlawing their use of the word "kosher" when issuing certificates to restaurants and caterers (HCJ 3336/04).

The many problems and irregularities involved in *kashrut* policy have led to the reform in the center of the current article, to which we now turn.

### 3.3. The Kashrut Reform

At the end of July 2021, the new *kashrut* plan of the Minister of Religious Affairs, Matan Kahana (of the Yamina party), was first unveiled. In October 2021 the reform was integrated into the existing *Kashrut* law.[15] The reform includes a number of changes. First, regional or municipal religious councils may certify *kashrut* in establishments outside their official jurisdiction areas (Rubin 2021). A similar system has been successful in improving service to couples registering for marriage by creating competition between religious councils (Ben Porat 2016), and should also enable a measure of choice within governmental systems in Kashrut regulation: businesses will be free to turn to the religious council who offers the best service.

Second, in the next step of the reform, which constitutes a groundbreaking change, private organizations will be allowed to supply *kashrut* supervision, if able to meet specified qualifications. The Chief Rabbinate will set fixed halakhic *kashrut* standards, as opposed to

the current situation where each local rabbi decides what these are. Private organizations will then be able to offer supervision in accord with the Rabbinate's requirements. Such organizations will be headed by a reputable rabbi, learned in the laws of *kashrut*, and will be supervised by the Rabbinate (Rubin 2021; Greenwood 2021).[16]

Third, three rabbis with proper credentials regarding their expertise in the laws of *kashrut* (at least one of which must be a certified city rabbi (*Rav Ir*), and two regional rabbis (*Rav Mo'atza*)), will also be able to regulate *kashrut* standards that may be either more lenient or more stringent than those dictated by the Rabbinate (Rubin 2021). The new private organizations will be able to choose which standard of *kashrut* to enforce: the one dictated by the Rabbinate, or one dictated by such a panel of rabbis (Houminer 2021).

Furthermore, organizations allowed to give *kashrut* certification in Israel will be able to award certification abroad. This element in the reform is aimed at lowering costs for those who import foodstuffs from aboard (Rubin 2021).

### 3.4. Responses to the Reform

The reform plan has been sharply criticized; mostly from the more traditionalist wing of Orthodox rabbis, as well as haredi members of Knesset and rabbis. The 'Kosharot' organization—a conservative and more stringent organization, which portrays itself as a *kashrut* authority—published a position paper claiming that the reform will be detrimental to the *kashrut* system in Israel (Kosharot 2021).[17] Additional rabbis such as R. Shmuel Eliyahu, the chief rabbi of Tzfat and a respectable halakhic authority (Eliyahu and Frank 2021), reputable haredi rabbis (Nachshoni 2022), and reactionary organizations such as Hotam (Hotam 2021) all criticized the reform, claiming it undermines the authority of the Rabbinate.

The opposition to the reform was also viewed as support for the rabbinic establishment, "protecting" it from the interference of the minister. This led to a smear campaign against the minister and rabbis supporting him and even to threats on the minister's life (Hauser-Tov 2021a, 2021b). Rabbis supporting the reform were also attacked, particularly R. Melamed, who is considered both a substantial halakhic authority as well as a stringent rabbi (see below).[18]

The Kohelet Forum, a think tank that describes itself as championing individual liberties and free market principles in Israel,[19] published a detailed review of the *kashrut* system and one of the Forum members, Amichai Filber, wrote an op-ed, criticizing the reform.[20] Filber posited that the new *kashrut* law does not allow for real competition, since it requires all *kashrut* bodies to act in accordance with a standard set by the Chief Rabbinate. Thus, with regard to raw materials, production processes, working methods, and supervision time—areas that each directly affect *kashrut* costs for each individual business—the Chief Rabbinate will remain the sole decisive body (Filber 2021).[21] It should be noted, however, that there is a possibility of circumventing the Rabbinate's standard using a standard set by three independent rabbis (see above); this option counters Filber's critique.

As a response to such critiques, Minister Kahana himself argued that he is acting with full rabbinical support. Among his supporters are the rabbis Re'em Ha'Cohen, Eliezer Melamed, Ya'acov Medan, David Stav, as well as others.[22]

At the same time, the ministry has also used economic framing when discussing the reform, presenting it as opening the market for competition that will benefit consumers (Ilan 2021b).[23] Interestingly, while the language of the law itself focuses on the economic aspect of the reform (as noted above), the minister (perhaps following the need to market the reform to more traditionalist constituents) also uses religious terminology when discussing the law and stresses how the reform will benefit the Rabbinate and bolster its power.

### 4. The *Kashrut* Reform and the Liberalization of the Israeli Economy

The *kashrut* reform is not a stand-alone policy. Rather, it is a part of a larger change transpiring in the Israeli society. Understanding this point requires an examination of the

changes the Israeli economy has undergone. Israel's early economic policies—dating from the country's establishment in 1948 to roughly the end of the 1970s—were characterized by large-scale governmental involvement in the market. Much of the country's GDP originated in governmental activities, a high percentage of workers participated in organized labor, and many limitations on monetary actions and capital flow were in place. Israel also offered its citizens the benefits of a welfare state, from universal health insurance to public universities. This quasi-socialist democratic model had its ideological roots in the ideology of the Labor party, Israel's dominant party from 1948 until 1977 (Ben-Bassat 2002).

Israel's initial economic approach began to change in the early to mid-1980s. This process is well documented (Zilberfarb 2005; Ram 2013), so a concise summary is sufficient here. The early 1980s were a period of economic crisis in Israel, with hyperinflation of over 400% annually. This crisis triggered a series of changes in the Israeli economy that continue to the present day. The main changes included a decrease in the power of labor unions, large-scale privatization, easing (or in some cases elimination) of constraints on economic activities of various kinds, a significant reduction in welfare policies, and more broadly speaking, the opening of Israeli markets to foreign capital and foreign workers. These economic changes are succinctly described by Zeira in his comprehensive research on the economy of Israel: "This shift was intense and widespread. It included cutting budgets, privatizing both state companies and public services, liberalizing trade, and weakening labor unions. We can therefore view it as a dramatic experiment in neoliberal economic policies" (Zeira 2021, p. 278).

These economic developments were accompanied by corresponding changes in many aspects of daily life in Israel, including the introduction of cable and satellite television, large-scale penetration of Internet connections, and the adoption of many Western trends, including fashion and music.[24]

During roughly the same period, several important changes occurred in the legal sphere, most notably the introduction of the Basic Law: Human Dignity and Liberty (1992), and the Basic Law: Freedom of Occupation (1992). These laws, accompanied by the interpretation of the Israeli Supreme Court, were considered a legal revolution by some (see the discussion in Barak-Erez 1994). The legal developments relevant to our discussion introduced (or expressed) an ideology of strong protection of property rights, individualism, and a general lack of intervention in the marketplace (Hirschl 1998).

The tensions between a policy of economic liberalization and local traditions that rely on statist protection are well documented in the literature on globalization (Castells 2011; Beyer 2007). While current versions of the study of globalization do not argue that globalization simply negates local cultures, globalization does expose local cultures, religions and even languages to pressures stemming from global aggregated preferences indifferent to the idiosyncrasies of localities. Furthermore, the individualistic nature of liberal ideologies can certainly bring about a perspective that would view the statist backing of such local cultures as limitations or impositions (Kymlicka 1995; Kukathas 2003).

It was perhaps inevitable that some religious attributes of the status-quo agreement would be seen as impositions by some, and indeed legal challenges to such Jewish features and laws presented themselves fairly quickly (Perez 2013; Ben-Porat 2013). An example for this scenario is commercial activity on the Jewish Sabbath (from Friday at sunset to Saturday night). According to Orthodox Judaism, commercial activity is prohibited during the Jewish Sabbath, and this view was reflected and maintained as a part of the status-quo agreement, with some exceptions. However, this has changed, and commercial activity during the Sabbath is currently (2022) widespread in Israel (Sapir 2018a). Interestingly, the legal framework regulating commercial activity during the Sabbath did not change in a country-wide fashion. Rather, the reality on the ground is the result of a somewhat odd combination of lack of enforcement, and a host of by-laws passed by different localities.[25] Be that as it may, the reality of commercial activity during the Jewish Sabbath at present is widespread in many locations across Israel, demonstrating a clear violation of the status-quo agreement.

Notably, religiously observant Orthodox Jews in Israel have, by and large, embraced capitalism, moving away from their traditionally socialist roots. This change, while not indicating secularization as atheism, does indicate the introduction of economic terms close to laissez faire ideology and the embrace of individual autonomy in the economic sphere; hence, many would also advocate reduction of governmental intervention and regulation as desirable policy attributes (Houminer 2017; Hellinger and Londin 2012). It is noteworthy, therefore, that the Minister for Religious Affairs who proposed the *kashrut* reform using economic language (illustrated above), is a religiously observant, Orthodox person.

Circling back to the *kashrut* reform, the change from a religious good provided by the state, to a state acting as a regulatory organ, can hence be seen as a part of the described overall change in the Israeli economy. This understanding leads us to the final section, attempting to redefine religion–state relations in Israel as a hybrid model.

## 5. Religion and State in Israel: The Rise of a Hybrid Model

The status quo model, as described above, has several components; the most important are (i) the religious monopoly over personal status, (ii) the statist provision of agreed-upon religious services and goods, and (iii) the exemptions and goods allocated to religious frameworks—most typically to Ultra-Orthodox institutions.[26] The laissez faire attributes of the Israeli economy influence all three parts, but especially the second one: the statist provision of religious goods. The *kashrut* reform is one important example, where the former policy was of direct provision of this good—via a state organ (the Chief Rabbinate— while the newly instituted policy is that the good will be provided via private bodies, and the state therefore retreats into a regulatory function. The magnitude of commercial activity on the Jewish Sabbath denotes even further movement, from a statewide ban to a de facto laissez faire situation, in which a decision whether to open a given business lies not with the state but with the owner of said business.

These developments arguably imply that the status-quo model is inadequate: it is unable to articulate central elements of religion–state relations in Israel. However, it is also impossible to argue that the status-quo model, as an institution regulating religion state relations in Israel, has become outdated by recent developments. As long as religious communities in Israel have exclusive authority over marriage and divorce, perhaps the most important aspect of the status quo model, the status quo model still maintains an important explanatory and descriptive capacity with regard to religion–state relations in Israel. However, the status quo model can no longer provide a proper explanation of the move to regulatory and laissez faire policies, as demonstrated in the *kashrut* and days of rest aspects. Notably, *kashrut* is one of the original and major aspects of the status-quo model; it is a sine qua non of Jewish life and of the status quo model (as explained in Section 3 above) and the changed policy is a major deviation from the status quo model. The growing reality of commercial activity during the Sabbath is likewise a major change; keeping the Sabbath is a major aspect of Jewish life and the status quo agreement (see Section 2 above), and the changed policies are too significant to simply be counted as minor deviations from the status-quo model.

We therefore suggest that the status-quo agreement no longer exclusively defines religion–state relations in Israel. Rather, religion–state relations in Israel in the 2020s are better defined when viewed as a hybrid model, drawing upon three different sources: the status-quo, libertarian, and regulation-based.[27] Such a hybrid model fulfills two functions: it is the model that regulates religion–state relations in Israel, and it is an explanatory model, aiming to denote, describe, and classify religion and state relations in Israel.

The hybrid model adds two additional attributes to the status-quo model. The first, the libertarian or the laissez faire aspect, is based on the assumption that statist intervention in individual decisions cannot be tolerated; individual autonomy should be preferred in all, or almost all, aspects of individual behavior. Individual autonomy includes property rights, leaving the state minimal functions only.[28] This view, having important economic

aspects (See: Hayek 1960; Friedman 1962), has adherents in Israel, especially, but not only in the economic sphere (See: Paz-Fuchs et al. 2018).

The second, the regulatory aspect, attempts to add complexity to the dichotomy between state and individual aggregated behavior. According to this school of thought, a policy of state retreat from a given sphere of activity is seldom, as an empirical matter, a full-fledged embodiment of laissez faire policy; rather it is replaced by "Governance through regulation (that is, via rule making and rule enforcement)" (Levi-Faur 2005, p. 13). As expected, the growing importance of regulatory frameworks can be seen in Israel as well (see: Tevet et al. 2020). A regulatory framework, briefly explained, is positioned between full statist provision of goods and pure policy of laissez faire. Regulatory frameworks imply that while the state (in many cases) will not provide a given good, private companies or NGOs wishing to provide it will have to meet numerous governmental sourced rules—this is a familiar aspect of many industries such as medical companies, construction sites, airlines, and so on. Importantly, a regulatory framework does not always mean that the relevant goods will be provided by a private organ; rather, the state might still provide it. However, the manner of manufacturing or importing/purchasing, distribution, allocation, and of course quality and safety and the like, will be regulated. Put succinctly, the presence of a regulatory framework substantially limits the autonomy of both the state and the relevant private organ in the provision of goods.

Bringing these two kinds of state–society frameworks together complements the status-quo framework: all three, put together, can adequately describe, explain, and classify different policies in religion state relations in Israel, where each category individually is unable to do so.

The status-quo model can explain the continued religious monopoly over marriage and divorce. If and when civil marriages are instituted in Israel, this will indeed mean the demise of the status-quo model. However, moving in this direction will require a major realignment of political forces, which is not currently in sight.

The laissez faire category can explain other parts of religion–state relations in Israel. The most adequate example is the state of affairs regarding commercial activity during the Jewish Sabbath. Here, the government employs a de facto policy of laissez faire—that is, it avoids enforcing the law, which is a part of the status-quo, which bans such activity, and moves the decision regarding commercial activity during the Sabbath to the sphere of decision of individual employees or business owners. Another example is the recent decision of the Israeli High court of justice, canceling a regulation that banned the bringing of *chametz* into state hospitals during the Passover holiday (HCJ, 1550/18 Secular Forum v. M. of Health).

The regulatory framework resides between full statist provision of goods and a laissez faire, hands-off approach. This view fits the new *kashrut* reform, in which the state, as of 2023, will cease to provide *kashrut* certificates, and will act as a regulatory body only.[29]

## 6. Conclusions

This article examined a major change in religion–state relations in Israel—the 2021 reform in *Kashrut* policy. The change is significant: from the beginning of 2023, the provision of *Kashrut* certificates will be provided by private bodies, while the state retreats to a regulatory function.[30] Together with the adoption of a de facto laissez faire policy in the domain of commercial activity during the Sabbath, we suggest that the status-quo model in religion–state relations in Israel can no longer be seen as defining religion–state relations, and is being replaced with a hybrid model, where the status-quo is complemented with laissez faire attributes and regulatory attributes. This hybrid model fits in with the changes in the Israeli economy, which has shifted from a centralized to liberalized economy. It would be interesting to see how the move to a regulatory function would influence the status and influence a state organ such as the Chief Rabbinate, and whether it would be able to commit to an environment of potential competition in standard setting.

A natural question concerns the next steps of this dynamic. Can the status-quo framework (even as one attribute within a hybrid model) withstand the advancement of the individualistic attributes of an economic liberalization? While a complete breakdown of the status-quo is not to be expected in the near future, the current reality hints at a gradual move from direct statist provision of religious goods to privatized and regulatory frameworks, applicable to various aspects of religion–state relations, to include domains such as public transportation during the Sabbath, introduction of welfare-to-work programs for Ultra-Orthodox Jews, regulating standardized curriculum for haredi schools, and many others. Put differently, we expect the economics of liberalism to gradually erode further aspects of the status-quo framework of the religion–state relations in Israel.

In a world where religion continues to be a viable political and social force, the Israeli case demonstrates that economic liberalization can shift behaviors and influence institutional reality of religion and state arrangements. This process can be slow and lengthy, but as the *kashrut* example demonstrates, it can and does happen. In this respect, the Israeli case arguably transcends this particular case study, as it demonstrates the radical changes that economic liberalism introduces to state-religion institutions, even in a case, like Israel, where the connection between religion and state is thought to be especially durable.

**Author Contributions:** Conceptualization, N.P. and E.R.; methodology, N.P. and E.R.; formal analysis, N.P. and E.R.; investigation, N.P. and E.R.; resources, N.P. and E.R.; data curation, N.P. and E.R.; writing—original draft preparation N.P. and E.R.; writing—review and editing. All authors have read and agreed to the published version of the manuscript.

**Funding:** This research received no external funding.

**Institutional Review Board Statement:** Not applicable.

**Informed Consent Statement:** Not applicable.

**Data Availability Statement:** Not applicable.

**Conflicts of Interest:** The authors declare no conflict of interest.

## Notes

[1] The *kashrut* reform has been linked to the reform in conversion, also initiated by Minister Kahana, and it may seem impossible to discuss one without the other. However, the conversion reform is still being debated and has yet to pass in the Knesset [Israeli parliament). We therefore limit ourselves to discussing the *kashrut* reform, which has already been formulated by law.

[2] Translated from the Hebrew by the authors, see p. 1165 of the economic plan (2021–2022), available online at: https://fs.knesset.gov.il/24/law/24_ls1_606661.pdf (accessed on 1 April 2022).

[3] Jewish thought on economic matters has a long history, but this historical narration is beyond the scope of the current article, see (Walzer et al. 2018, chps. 20, 21).

[4] In this we follow, *mutatis mutandis*, the school of thought that emphasizes the economic analysis of religious institutions, see (Iannaccone 1998).

[5] The topic of this article is the institutional arrangements between religion (Judaism) and state in the modern state of Israel; it of course touches upon Judaism as a religion, but that is not the main focus of the article.

[6] The Ottoman system was not egalitarian, being biased toward Islam, see Barkey and Gavrilis (2016).

[7] In 2010 a law was passed allowing 'civil union' for Israeli citizens who (both) do not belong to any recognized religion and wish to marry. This is however merely an ad hoc solution, as it introduces quasi marriage for a small number of Israelis. A majority of Israelis belong to a given religion, and religious belonging is assigned by the state upon birth; exiting such a belonging requires turning to the courts via an almost impossible process (see Supreme Court Case 7489/11, Kaniuk vs. the Minister of Interior).

[8] Another important example to the unusual autonomy of Ultra-Orthodox communities are private religious courts operated by various haredi communities, such as the Karelitz court in the mostly-haredi city of Bnei Brak. While most Haredi couples undergo divorce and marriage via the Rabbinate, they are able to change their personal status through these private courts. These courts are recognized by the State, in stark contrast to any other Jewish religious authority. See: Radzyner (2019).

[9] Both exemptions have long been accompenied by large scale social protest, and (especially the exemption from IDF service) also repeated litigation, and many attempts of various compromises, however, the concrete policy in both cases is, by and large unchanged; see (Sapir 2018b, chp. 8; Golan-Nadir 2022, chp. 4). For the current situation regarding the issue of haredim and the military see: (Rosman, forthcoming).

[10]　An important debate in this context is the attempt to allow graduates of the Ultra-Orthodox education system to study at colleges and universities in programs tailored to their level of knowledge, as to allow them to integrate to the Israeli labor market. Such programs are disputed as some of them attempt to duplicate the gender segregation attribute of Ultra-Orthodox education, a phenomenon that led to social and legal struggles that continue to the present (2022), see: Hartman and Zicherman (2019); Tirosh (2019).

[11]　Religious councils, operating at the municipal or regional level, regulate all issues related to supplying religious needs of a given Jewish community. This is an administrative body that also pays the salary of relevant (Jewish) clergy. See Jewish Religious Services law, 1971 (https://www.nevo.co.il/law_html/law01/p177_024.htm, accessed on 1 April 2022).

[12]　Some religious councils, such as Efrat and Acco, maintained a *kashrut* supervision system where the local religious council collected fees in an organized way from the establishments supervised and paid the supervisors; requiring them to punch a clock in order to make sure they were indeed supervising properly and ensuring supervision without corruption. The direct connection between the supervisor and the establishment being supervised was ruled as prohibited by the Supreme Court (HCJ 3336/04).

[13]　See *kashrut* fees as set in: Knesset Regulations 9829, 23 December 2021 (https://www.gov.il/BlobFolder/reports/fees_of_religious_councils/he/%D7%A7%D7%95%D7%91%D7%A5%20%D7%94%D7%AA%D7%A7%D7%A0%D7%95%D7%AA%209829%20%D7%A2%D7%9E%D7%95%D7%93%D7%99%D7%9D1263-1260%20%D7%90%D7%92%D7%A8%D7%95%D7%AA%20%D7%94%D7%9E%D7%95%D7%A2%D7%95%D7%AA%20%D7%94%D7%93%D7%AA%D7%99%D7%95%D7%AA%20%D7%9C%D7%A9%D7%A0%D7%AA%202022.pdf, accessed on 1 April 2022).

[14]　Tzohar is an organization of Orthodox rabbis who aim to make religious services more accessible to the general public. Its basic premise is that the Rabbinate creates many difficulties for the public and has a negative reputation that reflects badly on religion in general and Judaism in particular.

[15]　Jewish Religious Services Law (1971, last amendment: 18 November 2021) (https://main.knesset.gov.il/Activity/Legislation/Laws/Pages/LawPrimary.aspx?t=lawlaws&st=lawlaws&lawitemid=2001411, accessed on 1 April 2022).

[16]　The change in the Chief Rabbinate's function, from a direct provision of goods to a regulatory organ, does not necessarily mean diminuation in its status; as many regulatory organs are powerful actors in their respective spheres of activity (See, Sunstein and Thaler 2009).

[17]　Much of their criticism is unfounded. For example, they criticize the fact that the supervised establishment will pay supervisors directly, while this is the current situation. Under the terms of the reform, supervised establishments will pay a central agency—either belonging to a religious council or a private organization—which will pay supervisors a set salary. Likewise, their cost calculations are inaccurate.

[18]　See for example a pamphlet attacking R. Melamed personally: Ariel and Pinchas (2022). The pamphlet, 31 pages long, brings letters and articles attacking R. Melamed both directly and indirectly. Additionally, rabbis opposing the reform called for the banning of R. Melamed's books, which are widely used in religious communities as a halachic source. While this began due to R. Melamed's more accepting tone toward reform rabbis it was bolstered by the current reforms (Greenwood 2022).

[19]　See: https://kohelet.org.il/ (accessed on 1 April 2022).

[20]　Amichai Filber is also a lawyer, and headed the *kashrut* department at the Ministry for Religious Affairs in the past.

[21]　For more on Filber's position regarding the regulation of *kashrut* in Israel, see: (Filber 2018).

[22]　R. Melamed is usually considered the most stringent of these rabbis. He is also considered very learned (a *talmid chacham*) whose halakhic rulings are held in high regard in most non-haredi communities and it was surprising that he supported the reforms.

[23]　This ties in with Finke and Stark's (1988) work on religious economies and how pluralism in religious frameworks benefits believers. See also: Broyde and Zeligman (2021).

[24]　For an in-depth discussion of these cultural changes, and the shift from a more collectivist society to a more individualistic one, see: Rosman-Stollman and Israeli (2015).

[25]　For the legal situation, see: (Friedman 2019).

[26]　This is best seen in the autonomy of various ultra-orthodox organizations and systems, such as the haredi education system. That said, the state-sponsored religious education system also enjoys more generous funding than the general state-sponsored education system. See for example, (Ilan 2020).

[27]　The new hybrid model is different from the well known status quo model; importantly, we use the term 'status quo' as a specific term—denoting the religious-statist arrangement of the state of Israel, not in a general way denoting 'the existing state of affairs'. Furthermore, we do argue that a hybrid model, which still recognizes the status quo as one important attribute, adequatly captures the realities of religion–state arrangements in Israel, as long as religious communities have exclusive jurisdiction over personal law. As such it is distinct from simply an all encompassing neo-liberal order (on neo-liberalism see: Jessop 2012).

[28]　As van der Vossen describes: "Libertarians strongly value individual freedom and see this as justifying strong protections for individual freedom. Thus, libertarians insist that justice poses stringent limits to coercion." (van der Vossen 2019); See also (Nozick 1974).

[29]　In some cases, a combination of more than one aspect of the new, suggested hybrid model explains a given piece of religion-state relations policy. For example, the ritual baths policy involves all three components, see Perez and Rosman-Stollman (2019).

[30]    While it seems possible that the current (2022) government may not remain stable, the law itself has passed. Whether it will be fully implemented, remains to be seen.

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
