# Peer review of "From State Control to Regulation to Privatization of Religion–State Relations in Israel: Kashrut Reform as a Case-Study"

_religions, doi:10.3390/rel13050455_

Round 1
Reviewer 1 Report
1) The issue of deferring military service to the ultra-Orthodox men has nothing to do with the issue of the status quo. I suggest excluding it from the historical / theoretical review.
2) The argument that the status quo harms the Israeli economy is not convincing enough, and presenting the ultra-Orthodox autonomy as "problematic" is far-reaching. There are opposing views of this effect, and there are intra-Haredi processes that point to a growing Haredi integration.
3) The essence of the status quo is to maintain the status quo. If the hybrid model is implemented, it is not a "status quo" at all, but rather a new religious-state relationship.
4) It is important to clarify that the status of the Chief Rabbinate and the status quo will be harmed in the hybrid model, and therefore this process has implications beyond the economic aspect. Perhaps give another perspective that perceives religion-state relations also as having a value aspect and not merely an economic aspect.
5) In the new political circumstances (undermining the stability of the Bennett government) it is possible that the kashrut reform will not be implemented and will return to being a purely theoretical model. It is important to clarify this.
Author Response
Reviewer 1
The reviewer felt that the issue of deferring military service for ultra-Orthodox men has nothing to do with the issue of the status quo.
We have omitted this point from the discussion of the core principles of the status quo agreement. We still discuss it in the context of the model, since it is relevant to this issue.
The Reader felt that the argument that the status quo harms the Israeli economy is not convincing enough, and presenting the ultra-Orthodox autonomy as "problematic" is far-reaching. There are opposing views of this effect, and there are intra-Haredi processes that point to a growing Haredi integration.
Here we follow some of the most important economists who study the Israeli society, and their evaluation of the tense relations between the Ultra-Orthodox community and the education system and the Israeli economy, including Joseph Zeira (2021, 310-315) and Dan Ben David (2022) - See page 4 of the article.
The reviewer posited that the essence of the status quo is to maintain the status quo. If the hybrid model is implemented, it is not a "status quo" at all, but rather a new religious-state relationship.
Indeed, the new hybrid model is different from the status quo. We believe we say so explicitly. Also - we use the term ‘status quo’ as a specific term - denoting the religious-statist arrangement of the state of Israel, not in a general way denoting ‘the existing state of affairs’. See p. 10
The reviewer thought it important to clarify that “the status of the Chief Rabbinate and the status quo will be harmed in the hybrid model”, and therefore this process has implications beyond the economic aspect. “Perhaps give another perspective that perceives religion-state relations also as having a value aspect and not merely an economic aspect.”
Harmed is a value driven term; we are not evaluating the desirability of the change, we are describing it; we can simply state that the status of the Chief Rabbinate will change, from directly providing a good, to being a regulatory organ. There are many powerful regulatory organs (See Sunstein and Thaler, 2009), so the change in the Chief Rabbinate functions does not necessarily mean diminution in its status. See , p. 7, new f.n. 16.
The reviewer notes the new political circumstances (the stability of the Bennett government) that emerged after submitting the paper.
We have added a note (f.n.33) stating that due to the current political situation, it is possible that the kashrut reform will not be implemented, though the law itself has passed.
Reviewer 2 Report
The article deals with a new topic that has not yet been discussed in the academic research - the kashrut reform in Israel (2021), relying on an impressive bibliography, but in my opinion it emphasizes only one aspect of the discussion, which I am not convinced is the main aspect. The complete set of motives for enacting the law has not yet been revealed and its possible economic impact on food prices is in dispute. The reference to the reform as a violation of the status quo agreement is also problematic. The section dealing with kashrut, dealt with state kitchens, and not with the regulation of the kashrut standard. The ultra-Orthodox would have preferred their private courts to rule rather than the Chief Rabbinate. The reform directly affects the authority of the Chief Rabbinate and not the status quo agreement. In fact, in recent decades, following Israeli Supreme Court rulings, the status quo agreement and its subsequent laws (the Chametz Law, the Pig Law, the Hours of Work and Rest Law) have been eroded, making Israeli publicity more liberal and secular. This secularization find expression in four spheres: significantly increased commerce on the Sabbath, with large shopping centers opening on the outskirts of the cities; the import and sale of non-kosher meat; de facto recognition of civil marriages; and secular burials in private cemeteries. New religious legislation of essence, rather than of regulation, no longer exists. The Supreme Court has already intervened in the field of kashrut, and as a result the Chief Rabbinate itself has begun to change its policy and allow some degrees of kashrut, as well as issuing a kashrut certificate in hotels that place a Christmas tree, and this has had economic effects. I also do not think that kashrut reform can be discussed in isolation from the conversion reform, although it has not yet been approved. The kashrut reform is also not yet implemented in practice, so what remains are the motives for changes in regulation in both areas.
On page 8 it should be noted that Amichai Filber is a lawyer, who headed the kashrut department at the Ministry of Religions. Also, the attacks on Rabbi Melamed were for other reasons, and in some respects he is the least stringent of the others. He supports civil marriage, under certain conditions.
Author Response
Reviewer 2
Reviewer 2’s comments were written in paragraph form. We hence did our best to elucidated from her/his review the following points and suggestions.
We agree with Reviewer 2 that privatization has reached various aspects of religion-state regulation in Israel, for example in the opening of businesses in town centers during the Jewish Shabbat.
We also stress the connection between Israel’s economic liberalization and Israel’s religion-state framework.
We also agree with Reviewer 2 regarding the involvement of the HCJ in issues of Kashrut, as demonstrated by the recent decision regarding bringing Chametz into state hospitals during the Passover holiday – see p. 11.
Reviewer 2’s comment about state kitchens was less clear to us – as the reform regards the functionality of the Chief Rabbinate, and general standards of Kashrut, not only in state organs. This is explained in section 2.
We agree that the fact the conversion reform has yet to pass is important. We therefore discuss the kashrut reform, which has already passed into law, and only note the conversion reform in certain relevant places in the paper. We’ve added this emphasis in a footnote (f.n.1).
Reviewer 2 requested that we note that “Amichai Filber is a lawyer, who headed the kashrut department at the Ministry of Religions (p. 8)”.
This has been corrected.
Reviewer 2 notes that Rabbi Melamed is in some respects the least stringent of others. “He supports civil marriage, under certain conditions.”
On the scale between those who support the reform and those who oppose it, it is difficult to view R. Melamed as a “liberal” rabbi. As the reviewer notes, he does not “support civil marriage”, only accepts that at times it is a possible solution (see for example: Nahshoni, 2013 <https://www.ynet.co.il/articles/0,7340,L-4453706,00.html> ). Additionally, his support is for the Israeli option of spousal agreements/domestic partnership (Brit haZugiyut) only, and not state-sponsored civil marriage, to which he is still opposed.
The attack against him at present is directly connected to the conversion and kashrut reforms and his support of these. In the past R. Melamed opposed the kashrut reform (see: Melamed, 2015 <https://www.ynet.co.il/articles/0,7340,L-4453706,00.html> ). The current vehement attack began following his more accepting tone towards Reform Judaism, but was nevertheless bolstered by the developments surrounding the kashrut and conversion reforms (see: Greenwood, 2022 <https://www.israelhayom.co.il/judaism/judaism-news/article/7668536>). We therefore added a caveat regarding his change of attitude toward the Reform movement (f.n. 18), but respectfully request to stand by our own observation.
Reviewer 3 Report
I was hoping to learn more about Judaism as a religion and its role in politics, but I did not anticipate that while reading the article.
Author Response
Reviewer 3
The Reviewer was hoping to learn more about Judaism as a religion and its role in politics.
The topic of the article is the institutional arrangements between religion (Judaism) and state in the modern state of Israel; it of course touches upon Judaism as a religion, but that is not the main focus of the article. We have added a note in the introduction to the article clarifying this (f.n. 5).